# The Definition Dilemma: How Definitions of Disability Shape Statistics on Social Participation

**DOI:** 10.3390/ijerph22040603

**Published:** 2025-04-11

**Authors:** Judith Baart, Willem Elbers, Alice Schippers, Sarah Polack

**Affiliations:** 1Care Ethics, Disability Studies in Nederland, University of Humanistic Studies, 3512 HD Utrecht, The Netherlands; alice.schippers@disabilitystudies.nl; 2Gender & Diversity Studies, Radboud University, 6525 XZ Nijmegen, The Netherlands; willem.elbers@ru.nl; 3International Centre for Evidence in Disability, London School of Hygiene and Tropical Medicine, London WC1E 7HT, UK; sarah.polack@lshtm.ac.uk

**Keywords:** disability, Washington group questions, participation, UNCRPD, Sustainable Development Goals, inclusion, analysis

## Abstract

To monitor progress in including persons with disabilities, including in interventions which can improve their health and quality of life, it is crucial to collect data on their participation. However, there are many different ways of defining disability and thus categorising individuals as disabled/not disabled, which may impact measures of participation. This study aims to assess the relationship between three different measures of disability and the level of participation of persons with disabilities. We analysed data from a population-based survey of disability amongst Syrian refugees in Istanbul, which collected data on disability using the Washington Group enhanced set of disability questions as well a two-question set asking about perceived activity limitations due to disability. The prevalence of disability ranged from 13.5 to 22.4% depending on the measure used. The group of people that are categorised as disabled also differs, indicating who is being seen as disabled changes when a different measure is used. Levels of participation, with regards to paid work, education and being partnered, also varied by measure, for example, being in paid work ranged from 26% to 38%. These findings underscore the importance of carefully selecting and clearly defining disability measures in studies, (health) interventions and policy contexts.

## 1. Introduction

“We say that 16% of the world’s population has a disability. The majority of them do not use the word “disability” to describe themselves. To understand disability requires us to make space for diverse perspectives and diverse ways people describe their own experiences” [1].

The UN Convention on the Rights of Persons with Disabilities (UNCRPD) [2] and the Sustainable Development Goals [3], emphasise that persons with disabilities should have the same access to services and opportunities in society as those without. This means that the international development sector is expected to promote the inclusion of persons with disabilities in their policies and programmes, including in interventions aimed at benefiting health and quality of life. To understand the extent to which this is being achieved requires monitoring the participation of persons with disabilities. This requires understanding and defining who persons with disabilities are. However, different approaches are used to measure and define disability, which in turn are likely to influence the levels of participation that are measured and indicators of progress towards inclusion. This study aims to understand to what extent the level of participation of persons with disabilities varies depending on the measure of disability used in research.

The UNCRPD notes as its general principle “full and effective participation and inclusion in society” [2] (p. 5). The Sustainable Development Goals (SDGs) also highlight the importance of the inclusion of persons with disabilities, noting its goal to “empower and promote the social, economic and political inclusion of all, irrespective of age, sex, disability, race, ethnicity, origin, religion or economic or other status” [3] (p. 21). Inclusion is best defined as “a desired goal that requires equality of opportunity and participation in the rudimentary and fundamental functions of society” [4] (p. 35). Or, more commonly, “the opposite of social exclusion” (ibid). Policymakers need to collect data on persons with disabilities and monitor their participation to know whether they are achieving inclusion in their policies and programmes.

Societal participation of people with disabilities (i.e., are those with disabilities partaking and being served on an equal basis) is a key measure to track progress regarding the inclusion of persons with disabilities. As noted by the Washington Group on Disability Statistics, “If there are statistical differences in the number of people with disability in school or work, marrying or voting, then we can begin to discuss ‘exclusion’” [5] (p. 3). Participation of persons with disabilities and progress on the SDG targets can thus be assessed by disaggregating outcome indicators—such as employment, schooling, or access to health care. “These participation levels should be equal; those with disabilities should be participating equally in society compared to those without disabilities. If the levels are not equal, appropriate accommodations have not been made” [6] (p. 1169). This is similar to Rimmerman’s explanation of how social inclusion or exclusion is regularly measured, by “comparing the status of the excluded subpopulation with the general public by using multiple indicators” [4] (p. 42); comparing groups of people with and without disabilities can help us to “better understand the equalization of opportunity” [7] (p. 2).

Yet there is no universally agreed definition for disability: its meaning varies and is dependent on the context in which it is created [8,9,10,11,12]. Disability is complex, challenging and multi-dimensional, making it difficult to define, and the definition may need to change depending on its purpose [13]. The International Classification of Functioning, Disability and Health (the interaction between the underlying health condition/impairment and contextual factors) [14,15] developed by the WHO has, since its development, provided a common framework for conceptualising disability, but needs operationalising. Commonly used disability definitions result from different disability models, such as the functional (the inability to do something); the administrative (the individual has been processed as disabled by the welfare state); the subjective (the individual perceives him/herself as disabled); the social (barriers in society keep people with impairments from participating in society) [16] and the medical (disability is a deviation from the normal functioning of the body). More models and ensuing definitions exist globally, as ways of constructing disability come from the cultural and social context within which they exist [11].

The definition and subsequent assessment tools used to measure disability shapes who are labelled, categorised and recognised as having a disability. Our previous analysis comparing functional and subjective assessment tools of disability found that there was a core group of people who would be categorised as disabled using either assessment tool, but also people whose categorisation would depend on the tool used [17] (p. 10). Other studies have found poor agreement between functional assessments of disability (such as the Washington Group questions) with direct self-identification questions (such as “do you have a disability?”), with some studies finding less than 50% of respondents being categorised as “disabled” using both methods [7,17,18,19,20,21,22]. A comparison of a direct self-identification question of disability with administrative data (whereby students that had an Individual Education Plan were classified as disabled) found 89% of respondents being labelled as “disabled” using both methods, concluding that individuals considered the two concepts to be distinct constructs [23]. Studies comparing a clinical assessment of impairment to a self-reported functional measure also demonstrate differences in populations categorised as disabled [24,25,26]. One study, comparing four different disability measures found prevalence estimates ranging from 5.6% to 29.7% [27]. In summary, these studies indicate that “The different approaches taken […] result in the identification of substantially different, though overlapping, groups as disabled, and a range of different estimates of disability prevalence” [21] (p. 23).

Given that each method of measuring disability categorises different individuals as disabled, the process of choosing which method of categorisation is used is a political process that can potentially have far-reaching effects on the lives of individuals, such as whether or not they are eligible for social protection programmes. As noted by Beaudry, “once a definition of disability is used in legal contexts and social movements or embedded in a policy, it can do both harm and good. For instance, it can contribute to marginalising people with disabilities or it can empower them to make claims through their right to equality” [28] (p. 4). It is thus crucial to understand that labelling is a social construct dependent on who defines which labels are used and why; it is never apolitical [29].

We thus know from earlier research that the prevalence of disability is dependent on the measure of disability used: different tools identify different people as disabled. What has scarcely been studied yet and has not been studied in the international development context, is how differing disability measures would influence levels of participation, given that inclusion is assessed by comparing key outcomes of those with disabilities to those without disabilities. If participation measures change with changing disability assessment tools, then without thoughtfully addressing how disability is measured, and how individuals are categorised as disabled, information on the progress achieved in promoting inclusion may be skewed—as well as any assessments on the effectiveness of policies and programmes that target and promote inclusion.

This paper explores how the choice of disability measure influences findings on inclusion indicators of persons with disabilities, in this case in a refugee setting. Refugee populations offer a particularly relevant case for examining how different definitions of disability shape who is identified as disabled. Given their diverse experiences of displacement, trauma, and disrupted access to care, refugees are likely to present a wide range of functional limitations and psychological challenges. In such heterogeneous settings, the boundaries of disability are especially fluid and may shift significantly depending on the measurement tool used. This makes them a compelling population for exploring how definitional choices influence both categorisation and the way we assess inclusion or exclusion.

We compare three measures of disability: the Washington Group Short Set (WGSS), the Washington Group Short Set—Enhanced (WGE), which ask about the level of difficulty in different functional domains (e.g., seeing, hearing, self-care) and a two-question sequence asking about limitations due to disability or illness. The Washington Group questions and questions about limitations due to disability or illness are commonly used in censuses and surveys to generate disability statistics, but few studies have compared them, particularly within humanitarian contexts and conflict-affected populations. We assess the level of agreement and compare key indicators of inclusion (education, labour market participation and marital status) between people with and without disabilities categorised according to each of these disability measures.

## 2. Materials and Methods

For this study, we carried out secondary analysis using an existing data set from a population-based survey on disability amongst Syrian refugees in Sultanbeyli district in Istanbul in 2019, carried out by author S.P. and colleagues. Detailed methods are presented in a previous publication [30]. Briefly, multi-stage cluster sampling was used to select people for participation in the survey. Participants were visited in their households for an interview. In total, 4018 people aged 2+ years were enumerated, of which 3084 took part in the survey. Analysis for this paper was restricted to survey respondents aged 18 and older, leaving us with 1555 respondents to include in this study. This was because the Washington Group Questions used for children are different.

Data were collected on Android tablets with Open Data Kit software (Available online: https://getodk.org/ accessed on 7 April 2025). Ethical approval was obtained from Istanbul Sehir University, Republic of Turkey Ministry of Interior: Directorate General of Migration Management and the London School of Hygiene and Tropical Medicine. Informed consent was obtained from all participants. Those that were identified as having health needs were referred for further services. Data collected included socio-demographic and economic data including age, sex, education, marital status and employment.

For the purposes of the current analysis, we compared three measures of disability collected in the survey: the Washington Group Short Set (WGSS), the Washington Group Short Set—Enhanced (WGE) and a two-question sequence asking about limitations due to disability or illness, herein referred to as the “two-question set” (2QS) for brevity. Table 1 presents more details of each of these measures and the criteria used to classify people as disabled for the purposes of disability statistics.

We compared key indicators between persons with and persons without disabilities and assessed whether this differs depending on the measure of disability used. The measures of participation collected in this study were level of education, paid labour market participation and marital status. Analysis was completed using IBM SPSS Statistics 29. We calculated the prevalence of disability (with 95% confidence intervals) according to each measure, stratified by age and sex. The degree of overlap between each of the two WGQ sets with the 2QS was visualised using Venn diagrams. Odds ratios were calculated to compare participation indicators between persons with and without disabilities, with three separate analyses, one for each disability measure.

## 3. Results

### 3.1. Prevalence of Disability Measures

In total, 1555 respondents aged 18+ years were included in the analysis. Just over a third of the sample (40.5%) were aged 18–29 years old; only 5.1% were 60+ years old. Slightly more females (56.7%) than males (43.3%) were included in the sample.

Overall, 13.5% (95% CI 11.8–15.3; n = 210) were categorised as having a disability using the WGSS, 21.7% (95% CI 19.6–23.8; n = 337) using the WG-E and 22.4% (95%CI 20.4–24.6; n = 349) based on the 2QS. The number of people in the population that would be considered disabled thus increases considerably when the 2QS or WGE measure is used. The prevalence of disability increases with age, regardless of the measure used. Prevalence was higher for women using the Washington Group measures, and higher for men using the 2QS, but these differences were not statistically significant (Table 2).

### 3.2. Overlap and Differences Between Disability Measures

In Figure 1 and Figure 2 we present the overlap of the individuals categorised as disabled using the Washington Group measures and the individuals labelled as disabled using the 2QS measure. Figure 1 compares the WGSS with 2QS.

Looking at who makes up these groups: 4.7% of the full study population is categorised disabled using the WGSS but would be considered not disabled according to the 2QS measure, 13.6% are categorised disabled using the 2QS measure, but would be considered as not disabled according to the WGSS. Only 8.8% are categorised disabled by both measures, i.e., for disability statistics they would be considered a person with a disability regardless of the disability measurement used.

Comparing the WGE with the 2QS measure (Figure 2), 10.1% of the respondents are categorised as having a disability using the WGE but not the 2QS, 10.9% are categorised as having a disability using the 2QS but do not have a disability according to the WGE, 11.6% is categorised as having a disability using both the WGE and the 2QS measure.

Of all the respondents categorised as disabled using the WGSS or the WGE, older people are more likely to also be categorised as disabled using the 2QS (81.3% of 60+ year-olds would be considered disabled by both WGSS and 2QS; whereas only 39% of the 18–29-year-olds that are disabled using WGSS are also disabled using 2QS) (Table 3). That is, of the people who report difficulties in functioning, older people are more likely to also identify themselves as having a disability or illness that limits their activities.

### 3.3. Measures of Participation Using Different Definitions of Disability

Table 4 presents the three measures of participation, disaggregated by the three different measures of disability used in this study.

The percentage of respondents categorised as disabled who have not attended any schooling varies from 13% (2QS) to 14% (WGE) to 18% (WGSS). Regardless of the measure, the odds of having no formal education were significantly higher for people with disabilities; but the magnitude of the effect size varied slightly (WGSS: OR 2.53, 95% CI 1.69–3.78); 2QS: OR 1.82, 95% CI 1.26–2.63; WGE: OR 1.66, 95% CI 1.15–2.41).

Persons with disabilities were consistently less likely to be currently partnered (i.e., married or living together) than persons without disabilities, although this difference was only statistically significant using the WGE (OR 0.77 (95% CI 0.54–0.97).

For paid work (defined as having regular or irregular paid work or seeking paid work) we restricted the analysis to participants who identified as male, as very few women (<5%) in the study population were in paid work. Approximately a quarter (26%) of men with disabilities participate in paid work using the WGSS measure of disability, as compared to 35% using the 2QS and 38% using the WGE. Regardless of the measure, men with disabilities were significantly less likely than men without disabilities to be in paid work, but the magnitude of the difference was greatest when using the WGSS (OR 0.28, 95% CI 0.18–0.44).

## 4. Discussion

In order to understand whether the inclusion of persons with disabilities is being achieved, and to operationalise the ICF approach towards disability, requires the use of tools to measure disability in censuses and surveys. Using three different measures of disability on one population—in this case Syrian refugees in Turkey—enables us to analyse how research results change with varying definitions and ensuing measures of disability. First, we note that the disability prevalence in the population changes with each measure. This corroborates earlier research and general disability statistics [7,17,18,19,20,21,22,23]. Thus, any research, census or statistics that publish data on persons with disabilities should always note which measure and model of disability was used.

In addition, we see that the characteristics of those categorised as disabled change depending on the measure used: older people who are considered disabled using a functional measure of disability are more likely than younger people to also be considered disabled using a limitations measure of disability; those with mental health conditions such as anxiety or depression are more likely to be categorised as disabled using the WGES and the 2QS than the WGSS. People can thus experience disability differently based on different identities, such as age, sex or refugee status, and thus possibly also the intersection of multiple identities. Depending on the measure used, certain demographic groups may thus disproportionately be categorised as disabled or not disabled; this could exacerbate exclusion if these measures are used to decide on resource allocation for individuals, health interventions or as disability statistics to inform policy decisions.

We also see that the specific individuals who are categorised as disabled are not the same for each measure of disability. For example, while 8.8% of the population were categorised as disabled for both the WGSS and the 2QS measure, 4.7% were categorised as disabled using the WGSS but not the 2QS and 13.6% by the 2QS but not the WGSS. This aligns with earlier studies [17,25]. We conclude, as Beno Schraepen writes, that there is no one person with a disability but that different people have different characteristics that lead to exclusion or segregation [12] (p. 26). Each of the measures picks up people at greater risk of exclusion, but who the various measures categorise as being disabled deserves further attention.

Lastly, we see that, consequently, the conclusions drawn on the inclusion and participation of persons with disabilities slightly vary based on the disability measure used. In this study, we used no schooling, being married/living together and having paid work as our indicators of participation. For all outcomes, we see that there are inequalities between people with and people without disabilities. However, the magnitude of that inequality varies based on the disability measure used, reflecting, again, that each of the measures captures a different group of people. This is important, as such disaggregated data inform decision makers in developing interventions, policies and programmes.

To understand whether society, policies, programmes and interventions are inclusive of persons with disabilities, we need to demonstrate how persons with disabilities are doing in relation to their peers. This requires comparing persons with and without disabilities using the same indicators. However, using different definitions of disability categorises a different group of individuals as disabled: although the WGE and the 2QS find a similar prevalence of persons with disabilities in the population, the specific group that is categorised as disabled is not the same.

The findings demonstrate that there is no single, uniform category of “persons with disabilities” and that this has implications for subsequent outcomes on indicators of participation. Using different tools changes the extent of observed inequalities in key participation outcomes—in this case, education, paid work and partnership. This has significant implications, particularly when monitoring international frameworks such as the SDGs and the UNCRPD, whose aim is to achieve equality for those with and without disabilities. We need to acknowledge that “different methodologies lead to a range of understandings of disability” [32] (p. 5118). Further research is needed to understand who the people are that different measures of disability are categorising as disabled and how this is of relevance for policies, programmes, (health) interventions and the monitoring of the UNCRPD and SDGs.

Future research could delve into further examining how different measures of disability may impact the rights, social standing, participation, resource allocation and access to (disability-specific) resources of persons with disabilities. This involves conducting comparative studies using various disability measures to assess differences in access to legal protections, social services, and participation in societal activities.

It is important to recognise the diversity of disability and that the relationship between disability and participation will be shaped by intersecting characteristics and social identities. Data disaggregation by age, sex and type of impairment was not possible in our study due to limited sample size. However, a future analysis comparing measures should consider further disaggregation to provide a more in-depth understanding of these measures. Further research should also be conducted to analyse whether multiple measures of disability could be incorporated into data collection to capture a more comprehensive picture of the population that could be categorised as having a disability and are at risk of exclusion. This might be necessary to identify and address the needs of individuals who may be overlooked if only one measure of disability is used for categorisation, identification and resource allocation.

### Limitations

This study has several limitations. The measures of participation were restricted to schooling, marital status, and employment, which, while important, capture only a narrow subset of societal inclusion. Further, although employment is an indicator of current participation, for education it is not known whether the onset of disability was before, during or after school age. Broader dimensions of participation, such as social connectedness, access to healthcare, and civic engagement, were not collected on the full study population and therefore could not be included in this analysis. This deserves attention in future research as these additional indicators could provide a more nuanced understanding of the barriers and opportunities experienced by persons with disabilities, particularly in contexts of displacement. The prevalence of mental health conditions, including anxiety and depression, within the refugee population studied may have influenced how individuals perceived and reported physical and sensory functioning. This could result in an overestimation of difficulties, particularly in a population already exposed to significant stressors and trauma (see [30] for more information). This study provides evidence that participation measures differ depending on the measure of disability used. It would be of interest to replicate this amongst a non-humanitarian population.

## 5. Conclusions

The findings of this study underscore the importance of carefully selecting and clearly defining disability measures in both research and policy contexts, as it impacts the levels of participation measured when studying the inclusion of persons with disabilities. Since definitions can both marginalise and empower individuals, it is crucial to understand and communicate the implications of the chosen measures. There is no “gold standard” to measure disability, but the most appropriate measure would be dependent on the intended purpose with eligibility for, for example, health screening programmes, requiring a different measure than monitoring the participation of those at risk of exclusion in social protection policies. Studies and statistics on disability should therefore always include the specific measure or definition used.

## Figures and Tables

**Figure 1 ijerph-22-00603-f001:**
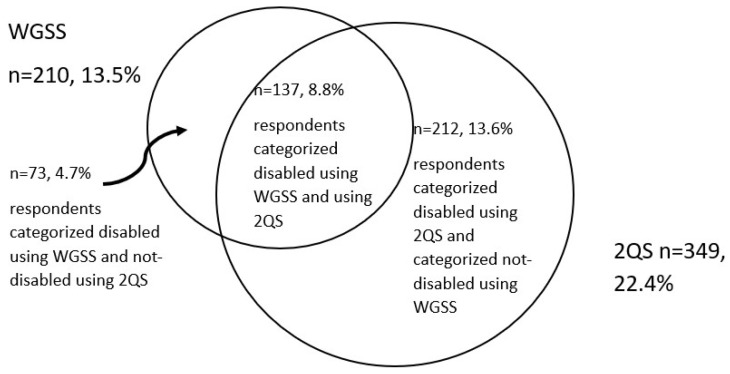
Percentage of respondents categorised as disabled by 2QS and WGSS.

**Figure 2 ijerph-22-00603-f002:**
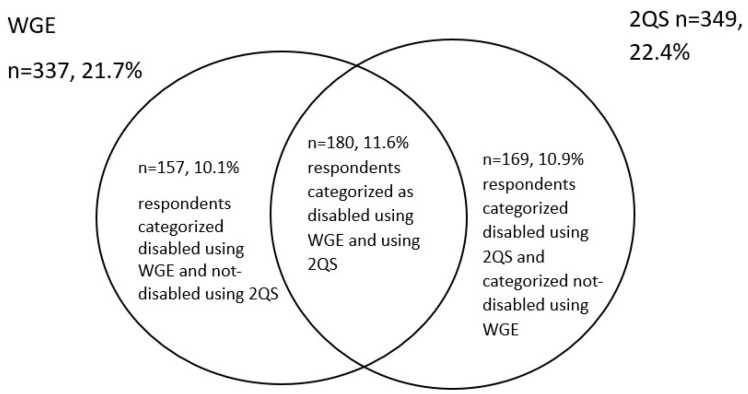
Percentage of respondents categorised as disabled by 2QS and WGE.

**Table 1 ijerph-22-00603-t001:** Disability measures used in the research.

Disability Measures	Notes	Questions	Response Options	Disabled If…
Washington Group Short Set (WGSS)	• Based on functional model of disability; • Works within ICF framework; • Focuses on activity limitations; • The word “disability” is not used during questioning to avoid associated stigma.	1. Do you have difficulty seeing even when wearing your glasses? 2. Do you have difficulty hearing even if using a hearing aid? 3. Do you have difficulty walking or climbing steps? 4. Do you have difficulty remembering or concentrating? 5. Do you have difficult (with self-care such as) washing all over or dressing? 6. Using your usual (customary) language, do you have difficulty communicating, for example understanding or being understood?	1. No difficulty. 2. Some difficulty. 3. A lot of difficulty. 4. Cannot do at all.	Categorised as Disabled if responded ‘a lot of difficulty’ or ’cannot do at all’ for any of the six questions.
Washington Group SS—Enhanced (WGE)	• Starts with WGSS; • Adds additional questions to include more functional domains (upper body functioning, anxiety and depression).	Includes the six questions as above, as well as: 7. Do you have difficulty raising a 2 litre bottle of water or soda from waist to eye level? 8. Do you have difficulty using your hands and fingers, such as picking up small objects, for example a button or pencil, or opening or closing containers or bottles? 9. How often do you feel worried, nervous or anxious? 10. Thinking about the last time you felt worried, nervous or anxious, how would you describe the level of those feelings? 11. How often do you feel depressed? 12. Thinking about the last time you felt depressed, how depressed did you feel?	For 7 and 8, same as above. For 9 and 11: 1. Daily. 2. Weekly. 3. Monthly. 4. A few times a year. 5. Never. For 10 and 12: 1. A little. 2. A lot. 3. Between a little and a lot.	Categorised as Disabled if at least 1 domain/question is coded ‘a lot of difficulty’ or ‘cannot do at all’ for the six short set questions, severity level 3 or 4 for Upper body Indicators, and severity level 4 for Anxiety–Depression Indicators.
Two question set (2Qs)	• Subjective measure of disability; • Used in UK census [31] • Asks whether respondents feel limited due to illness or disability.	1. Do you have any long-standing illness, disability or infirmity? 2. Does this limit your activities in any way?	1. Yes. 2. No.	Categorised as Disabled if YES to both questions.

**Table 2 ijerph-22-00603-t002:** Prevalence of disability in the population: WG short set, WG Enhanced and 2QS, by sex and age group.

	WG Short Set		WG—Enhanced		2QS		Total
	N	% (95% CI)	N	% (95% CI)	N	% (95% CI)	N
**Full Sample**	210	13.5 (11.8–15.3)	337	21.7 (19.6–23.8)	349	22.4 (20.4–24.6)	1555
**Age**							
18–29	41	6.5 (4.7–8.7)	85	13.5 (10.9–16.4)	60	9.5 (7.3–12.1)	630
30–39	50	11.2 (8.5–14.5)	89	20 (16.4–24.0)	95	21.3 (17.6–25.5)	445
40–49	44	18.4 (13.7–23.9)	68	28.5 (22.8–34.6)	86	36.0 (29.9–42.4)	239
50–59	43	26.7 (20.1–34.2)	60	37.3 (29.8–45.2)	67	41.6 (33.9–49.6)	161
60+	32	40.0 (29.2–51.6)	35	43.8 (32.7–55.3)	41	51.3 (39.8–62.6)	80
**Sex**							
Male	83	12.3 (9.9–15.1)	129	19.2 (16.3–22.3)	161	23.9 (20.7–27.3)	673
Female	127	14.4 (12.2–16.9)	208	23.6 (20.8–26.6)	188	21.3 (18.7–24.2)	881

**Table 3 ijerph-22-00603-t003:** Demographic characteristics of persons categorised as disabled by both WGSS or WGE and 2QS.

	WGSS			WGE		
	Total Categorised as Having a Disability by WGSS	Also Categorised as Having a Disability According to 2QS		Total Categorised as Having a Disability According WGE	Also Categorised as Having a Disability According to 2QS	
	N	N	%	N	N	%
**Age**						
18–29	41	16	39.0	85	26	30.6
30–39	50	31	62.0	89	48	53.9
40–49	44	32	72.7	68	40	58.8
50–59	43	32	74.4	60	38	63.3
60+	32	26	81.3	35	28	80.0
**Sex**						
Male	83	58	70.0	129	74	57.4
Female	127	79	62.2	208	106	51.0
**Total**	210	137	65.2	337	180	53.4

**Table 4 ijerph-22-00603-t004:** Percentage and odds ratios of not having had any schooling, being married/living together and doing paid work, by different measures of disability.

Participation Indicator	Disability Measure	Disabled N (%)	Not-Disabled N(%)	OR (95% CI)
**No formal education**	WGSS	38 (18%)	109 (8%)	2.53 [1.69–3.78]
	WGE	47 (14%)	100 (8%)	1.82 [1.26–2.63]
	2QS	46 (13%)	101 (8%)	1.66 [1.15–2.41]
**Married/living together**	WGSS	162 (78%)	1088 (82%)	0.79 [0.55–1.13]
	WGE	258 (77%)	992 (82%)	0.77 [0.54–0.97]
	2QS	280 (81%)	970 (81%)	0.94 [0.70–1.28]
**Paid work (men only)**	WGSS	21 (26%)	396 (68%)	0.28 [0.18–0.44]
	WGE	48 (38%)	369 (69%)	0.43 [0.32–0.58]
	2QS	56 (35%)	361 (72%)	0.44 [0.33–0.60]

## Data Availability

Data cannot be made publicly available as consent foe this was not obtained from study participants. For more information about the data, please contact Sarah Polack at sarah.polack@lshtm.ac.uk.

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
