# Peer review of "The Definition Dilemma: How Definitions of Disability Shape Statistics on Social Participation"

_ijerph, 2025, doi:10.3390/ijerph22040603_

Round 1
Reviewer 1 Report
Comments and Suggestions for Authors
Please check the attached file

Author Response
Dear reviewer,
Thank you for taking the time to review our article and helping us improve upon it. Please find in the attachment our response to your feedback.
Kind regards,
the authors

Reviewer 2 Report
Comments and Suggestions for Authors
Dear all, I would like to thank you for the opportunity to review an article with such an interesting theme. In fact, by adopting different definitions of disability, we will have different implications in different fields.
In general, the text is well written and the results are well presented and in line with the objectives. Here are some observations about the paper.
#1 The paragraph from line 47 to 52 would be better placed in the methods section and it would not be recommended that it appear in the introduction;
#2 There is a central issue in the manuscript presented regarding the failure to present a definition of participation that conceptually anchors the data analysis. According to the International Classification of Functioning, Disability and Health - ICF, participation is defined as “involvement in a life situation” and is also influenced by environmental factors. Almost all of the indicators used in the article (level of education, paid labor market participation and marital status) would not fit into the definition of participation presented. Therefore, for the article to have its argument maintained in a logical manner, it is necessary for the authors to present a definition of participation aligned with all of the indicators used.
#3 The WHO, through the ICF, also proposes different definitions of disability and participation. It is understandable that this approach was not used in the data collection, probably for logistical reasons. However, it is unacceptable that these definitions are ignored in the discussion of the article. The approach and definitions presented in the ICF may be recommended or contraindicated by the authors in the discussion of the article, but this information must necessarily dialogue with the data. In the ICF, WHO presents a definition of disability and participation that is recommended worldwide; not considering these definitions in the discussion is missing the opportunity for criticism and scientific debate that leverages science.
#4 Many elements that are in the conclusions could appear in the discussion, making the last section more concise.
Author Response
Dear reviewer,
Thank you very much for taking the time to review our article and helping us improve upon it. Please find our response to your feedback in the attached document.
Kind regards
the authors

Round 2
Reviewer 1 Report
Comments and Suggestions for Authors
Dear. authors
I appreciate the authors’ efforts to improve the manuscript.
Although some modifications were made on the manuscript, it is still unclear why the authors focused on the refugee population.
The authors described in the revised manuscript that “few studies have compared them, particularly within humanitarian contexts and conflict-affected populations.” This added sentence does not provide enough justification for focusing on the refugees. I think the authors should provide more clearer and more specific explanations why it is important to focus on the refugee populations. Moreover, it is also unclear how do the authors want to utilize the knowledge based on the current study findings. Tofani M and colleagues (PMID: 36292309) conducted a similar study focusing on migrants in Italy with clear explanation why they focused on this population and clear implications for practices based on their study findings. This previous study may be a reference to improve and refine the current study.
Furthermore, while the authors described as “What is not yet known is whether differing disability measures would influence levels of participation,” it has been reported that different disability measurement led to different results on social participation (e.g. Weeks JD et al. (PMID: 34546873)). Therefore, this added sentence does not provide clear explanation on the novelty for this study.
These two issues (unclear reasons for focusing on the Syrian refugee and the novelty of this study), which are important bases for the current study, are not satisfactory addressed in the revised manuscripts. I recommend authors to reconsider these issues (e.g., research purpose and its rational, novelty of research and potential implication of the research findings) and refine the manuscript.
END
Comments on the Quality of English LanguageNone.
Author Response
Dear reviewer,
Thank you for your feedback which has helped us improve the manuscript. We have aimed to address the two issues, which we believe have indeed now made the text more readable and clear. We have done so by reorganizing the introduction to make the argument more logical, and emphasizing further in lines 114-133. We have reorganized the conclusion to emphasize our main addition to the body of knowledge, as well as added the research implications in line 299-302.
We believe these additions further explicate the novelty and added value of this study, and look forward to any additional improvements you might recommend to improve the article.
Kind regards
the authors
Reviewer 2 Report
Comments and Suggestions for Authors
Dear authors, I appreciate the answers provided, although I do not agree with them, I recognize that this is a personal point and discussion would not be productive at this time. Therefore, I have no further comments on the manuscript.
Author Response
Dear reviewer,
Thank you for your response and appreciation. We appreciate the time you have spent on reviewing and improving the article.
Kind regards
the authors